# Friend or Foe? Exploring the Role of Cytomegalovirus (HCMV) Infection in Head and Neck Tumors

**DOI:** 10.3390/biomedicines12040872

**Published:** 2024-04-15

**Authors:** Aleksandar Trivic, Jovica Milovanovic, Djurdjina Kablar, Ana Tomic, Miljan Folic, Ana Jotic, Nada Tomanovic, Ana Marija Tomic, Igor Djoric, Marko Jankovic

**Affiliations:** 1Clinic for Otorhinolaryngology and Maxillofacial Surgery, University Clinical Center of Serbia, 2 Pasterova Street, 11000 Belgrade, Serbia; drcole71.at@gmail.com (A.T.); jovica.milovanovic@med.bg.ac.rs (J.M.); mfolic@yahoo.com (M.F.); anajotic@yahoo.com (A.J.); 2Faculty of Medicine, University of Belgrade, 11000 Belgrade, Serbia; nada.tomanovic@med.bg.ac.rs (N.T.); anamarija.tomic1@gmail.com (A.M.T.); igidren@gmail.com (I.D.); 3Department for Pathology, Pathohistology and Medical Cytology, University Clinical Centre of Serbia, 11000 Belgrade, Serbia; kablar.djurdjina@gmail.com; 4Center for Radiology Imaging, University Clinical Center of Serbia, 2 Pasterova Street, 11000 Belgrade, Serbia; anatomic9977@gmail.com; 5Institute of Pathology, 1 Dr. Subotica Street, 11000 Belgrade, Serbia; 6Clinic of Neurosurgery, University Clinical Center of Serbia, Institute of Radiology, 4 Dr. Koste Todorovića Street, 11000 Belgrade, Serbia; 7Department of Virology, Institute of Microbiology and Immunology, 1 Dr. Subotica Street, 11000 Belgrade, Serbia

**Keywords:** cytomegalovirus, cancer, head and neck tumors, oncoprotective, global

## Abstract

Although not regarded as an oncogenic pathogen, the human cytomegalovirus (HCMV) has been associated with a wide array of malignancies. Conversely, a number of studies report on possible anti-tumor properties of the virus, apparently mediated via HCMV-galvanized T-cell tumor killing; these were recently being investigated in clinical trials for the purposes of anti-cancer treatment by means of dendritic cell vaccines and HCMV-specific cytotoxic T cells. In the present study, we have analyzed the relation between a complement of head-and-neck tumors and HCMV infection across 73 countries worldwide using Spearman correlation, univariate and multivariate regression analysis. Intriguingly, HCMV was found to be pro-oncogenic in patients with nasopharyngeal carcinoma; contrarywise, the virus manifested an inverse (i.e., anti-tumor) association with the tumors of the lip/oral region and the salivary glands. Although this putative protective effect was noted initially for thyroid neoplasia and hypopharyngeal tumors as well, after multivariate regression analysis the connection did not hold. There was no association between laryngeal cancer and HCMV infection. It would appear that, depending on the tissue, HCMV may exert both protective and oncogenic effects. The globally observed protective feature of the virus could potentially be utilized in future therapeutic approaches for salivary tumors and neoplasia in the lip/oral region. As correlation does not necessarily imply causation, more in-depth molecular analyses from comprehensive clinical studies are warranted to substantiate our findings.

## 1. Introduction

The human cytomegalovirus (HCMV) is a member of the *Orthoherpesviridae* family, a well-known group of pathogens causing persistent infections in hosts. HCMV is a widespread and prevalent infection, reaching nearly 100% pervasiveness in some regions [1]. While generally considered harmless in healthy individuals, HCMV can pose a significant threat and lead to fatal outcomes in immunocompromised individuals.

The interaction between HCMV and its host is intricate and multifaceted, with extensive debates surrounding the virus’s potential oncogenic capacity. Cytomegalovirus has been linked to various malignancies, including brain tumors, breast cancer, and hematopoietic cell lineage neoplasms [2,3,4,5]. Numerous factors suggest a pro-oncogenic role or, at the very least, oncomodulation by the virus [6]. Significant evidence speaking in favor of a purported oncogenic effect stems from studies that reported on the presence of the virus in many cancer histology types; this has been extensively investigated in works that explore this relation between HCMV and glioblastoma multiforme (GBM), one of the most malignant tumors known [7].

Conversely, some studies indicate an anti-tumor effect of HCMV [6,8,9,10]; in recent years, accumulating evidence supports the notion of a T-cell mediated host anti-cancer response targeting HCMV molecules expressed on transformed cells. Recent data from clinical studies as well speak in favor of an oncopreventive role of HCMV, as demonstrated in studies of patients with colon, lung and B-cell malignancies [11,12,13]. However, the duality of the role of HCMV in cancers continues to persevere.

The role of HCMV in tumors, if any, has so far been explored in the tumors of the head and neck (H&N) region only seldom; for some malignancies, there is even no data at all. Furthermore, all of the studies are single-center investigations, and a global outlook on the issue is lacking. The arguments supporting the potential oncoprotective role of HCMV revolve around the proposed T-cell mediated elimination of tumors, bolstered by HCMV antigens’ potent immune-boosting properties. This theory gains credence from the observed absence of tumor-killing activity in the T-cell deficient milieu of Kaposi’s sarcoma/AIDS [14]. If the supposition holds true, we theorize that this oncoprotective capacity of HCMV may manifest in tumors of the H&N region. In our research, we inquired into the relationship between HCMV and a complement of tumors from the H&N anatomical locale as reported by the World Health Organization (WHO). Finally, we scoured the literature for works pertaining to the link between these cancer types and HCMV in order to give a comprehensive viewpoint on etiology of HCMV in cancer.

## 2. Materials and Methods

In the method of choosing the relevant tumors of the H&N region (as classified by the World Health Organization [15]), we have deferred to the data gathered by the Global Cancer Observatory (GLOBOCAN), a division of the WHO; GLOBOCAN keeps with one of the primary aims of the International Agency for Research on Cancer (IARC)—to describe and elucidate cancer occurrence worldwide [16]. 

To investigate the interplay between HCMV and H&N region malignancies, we have explored the correlation between country-specific age-standardized cancer incidence rates and corresponding HCMV seroprevalences. To account for potential confounding variables, specifically those that could impact the prevalence of HCMV, we utilized multivariate logistic regression (MLR).

The cancer types/categories analyzed in this study, in line with the nomenclature used by GLOBOCAN, were as follows: “thyroid”, “salivary glands”, “oropharynx”, “nasopharynx”, “lip/oral”, “larynx” and “hypopharynx”. Additionally, we have included the “all cancers” category in order to compare the results with the incidence of all cancers combined.

The age-adjusted annual incidence rates, reported per 100,000 individuals, have been documented for 7 cancer categories in 185 countries using data from GLOBOCAN. Incidences were collectively observed for both males and females, covering the entire specified age range (0–85+ years).

The prevalence of HCMV was illustrated using country-specific viral seroprevalence data for a total of 73 countries. These data were compiled by Zuhair et al. [1], who conducted a systematic survey of published literature to provide perspectives into the global prevalence of HCMV IgG antibodies. 

A number of variables may influence HCMV pervasiveness. The selection of these confounding factors was influenced by their documented correlation with the prevalence of HCMV in the literature: for a parallel of socioeconomic status, the human development index (HDI) was used [17,18,19], average population age [20,21,22], estimated number of sexual partners [17], smoking prevalence (used as a substitute for current smoking) [17], and the percentage of children born in the last 2 years who were ever breastfed [21]. The information used for these variables was accessed from the United Nations Development Programme 2021 Human development reports [23], United Nations World Population Prospects (2022) [24], World Population Review website [25], Tobacco Atlas (University of Illinois, WHO GTCR 2023 data) [26], and UNICEF [27], respectively. For the assessment of the country income level, data from the World Bank were used [28].

The search for papers was conducted on the PubMed^®^ platform in January 2024. The details of the search methodology, results, literature, and other relevant data can be found in Table 1, while the list of examined malignancies is presented in Table 2.

### Statistical Analysis 

Statistical investigation, i.e., comparison between age-standardized annual cancer incidence rates and country-specific HCMV seroprevalence was performed via means of the Spearman’s rank correlation test, which was complemented regression analyses (Appendix A). Univariate and multivariate linear regression analysis was performed for every selected tumor type, using HCMV as an independent predictor in univariate analysis—ULR (based on correlation analysis results as well as expert opinion) and further adjusted for confounding factors in multivariate analysis. Confounding factors for multivariate regression were chosen based on their literature-based association with HCMV prevalence, with human development index (HDI) used as a parallel of socioeconomic status, encompassing three major factors: life expectancy, education (mean years of schooling completed and expected years of schooling upon entering the education system), and per capita income, along with cofounders such as country specific average population age, smoking, and breastfeeding. The analyses were carried out in SPSS v.26 (IBM corp., New York City, NY, USA) statistical software. *p*-values were used to denote corresponding levels of statistical significance.

## 3. Results

An inverse Spearman’s correlation was evident between the prevalence of HCMV and all of the estimated age-adjusted tumor incidence rates for all of the analyzed tumors from the H&N region. A high degree of statistical significance (*p* < 0.001) was demonstrated in all of the cases (Table 2). It is worthwhile noting that this statistical association persisted when considering incidence rates for all cancers combined (Spearman’s ρ = −0.732, *p* < 0.001). 

As presented in Table 2, an analysis of univariate linear regression (ULR) was conducted, with HCMV being used as an independent variable. The prevalence of HCMV is a substantial and independent predictor of certain tumor-type incidence, strongly and highly correlated with a change in the occurrence of noted tumors. Upon investigating the influence of HDI as a confounding factor in multiple linear regression analysis, it was observed that HCMV remained a strong predictor for a reduced occurrence of tested malignancies, exhibiting favorable overall MLR model characteristics.

Negative Spearman’s ρ and Standardized coefficients β from ULR and MLR suggest that HCMV provides oncoprotection in malignancies of the oropharynx, as well as tumors of the lip/oral cavity and salivary glands, as increased HCMV prevalence is associated with decreased tumor incidence, where significant. 

After adjusting for country specific HDI, MLR analysis revealed no significant link between thyroid, hypopharynx, and larynx occurrence.

On the contrary, HCMV incidence was perceived as an independent predictor for tumors of the nasopharynx, maintaining its significance as a cancer inducing agent factor even after adjusting the analysis for HDL. 

After accounting for additional confounding factors such as age, smoking, and breastfeeding, the findings of the MLR analysis did not show any significant differences. Therefore, they were not included in the final model. The visualization of the Spearman correlations concerning HCMV prevalence vis-à-vis tumor incidences is presented in Figure 1A–G.

## 4. Discussion

To date, there have been numerous studies both supporting and questioning the concept of HCMV oncogenicity. Despite not being officially recognized as an oncogenic pathogen, the human cytomegalovirus has been linked to a diverse range of malignancies. Conversely, mounting evidence speaks in favor of potential anti-tumor properties of the virus, making this discussion a polarized one. The investigation of HCMV’s role in ENT cancers of the H&N region has been a very limited one, lacking an in-depth worldwide perspective on the issue. In this study, we aimed to provide a more comprehensive understanding of the correlation between this virus and specific H&N tumors on a global scale. We share findings from our research and incorporate results from other studies to provide a clearer and unbiased perspective on the potential relationship between HCMV and tumors—if any at all.

In this study, statistical analysis affirms that HCMV stands as the only significant factor for oncoprotection in salivary gland, lip/oral cavity, and oropharyngeal tumors. Interestingly, the human development index (HDI) does not appear to contribute to the variation in tumor incidence. More than a century ago, there was speculation about the potential of certain viruses to induce tumor regression and remission [52]. It was later discovered that these viruses exhibit a natural affinity for cancer cells, leading to the destruction of infected cells and triggering host anti-cancer immunological responses [53]. Moreover, oncolytic viruses designed to lyse tumor cells have been identified among coxsackievirus, adenovirus, and herpes simplex virus strains [54]. In recent times, there has been growing interest in considering HCMV as a potential safeguard against cancer. Patients with B-cell malignancies were found to exhibit a notably lower incidence of HCMV seropositivity when compared to the control group [13]. Additionally, a clinical study by Nagel et al. [11] highlighted a significant correlation between HCMV infection and a lessened incidence of colorectal carcinoma. Viral infection was associated with decreased Risk of Bronchogenic Carcinoma in another investigation, underscoring the need for further investigation into how the tumor microenvironment and host immune system are altered by the presence of a latent HCMV infection [12]. 

Indirect evidence, and admittedly not definitive, constrains the association between HCMV and tumorigenesis, namely, HCMV DNA is notably absent in investigations of pleomorphic adenomas [55], Warthin’s tumors [55], epithelial ovarian cancer [56], papillary thyroid cancer [41], and pediatric medulloblastomas [57].

Geris et al. determined that the HCMV status did not exhibit a correlation with the risk of secondary cancer development in SOT recipients [58]. Moreover, these authors acknowledged a reverse correlation between HCMV and diffuse large B-cell lymphoma (DLBCL), aligning with findings from our previous study [13]. A similar result was observed in an experimental model with murine HCMV, where the virus negatively impacted the progression of B-cell lymphoma [59].

Other studies with transplanted patients yield evidence for a virus vs. tumor effect by HCMV. In individuals who have undergone allogeneic hematopoietic cell transplantation (HCT), HCMV reactivation has been linked to a noticeable reduction in the risk of leukemia relapse [60]. This finding is supported by the observation that swift HCMV replication may lower the risk of relapse in non-Hodgkin lymphoma [61], acute myeloid leukemia [62,63,64], and pediatric acute leukemia post-HCT [65]. HCMV reactivation following HCT was associated with a slight decrease in the risk of early relapse in a cohort of patients with myeloproliferative disorders [66].

In malignant tumors of the lip and oral cavity, our study suggests an anti-tumor effect of HCMV. While no specific research on the association between HCMV and oral malignancies was identified, Mattila et al. documented a case of lip cancer developing in an immunocompromised HCMV-infected patient [67]. Saravani et al. reported HCMV presence in a minor number of subjects with oral squamous cell carcinoma, without establishing a clear cause and effect relationship [50]. Similarly, studies from Taiwan [68] and Pakistan failed to find a definitive link between HCMV infection and oral malignancies [51].

Our findings suggest an oncoprotective effect of HCMV in salivary gland tumors. Melnick et al. investigated salivary gland pathology [45], as well as reported active HCMV infection in a majority of mucoepidermoid carcinoma subjects, contrasting with Jayaraj et al., who found no evidence of HCMV using immunohistochemical staining, possibly due to methodological differences [46,69]. In a 2017 study by Chen et al. [70], the presence of HCMV in malignant tumors of the salivary gland was detected by the PCR method, while Radunović et al [48]. found HCMV infection using immunohistochemical analysis in a significant percentage of patients, though they also observed HCMV in normal salivary gland ducts. On the other hand, Atula et al. (1998) and Kärjä et al. (1997) in their respective inquiries found no association between HCMV and malignant salivary gland tumors [46,47].

The initially observed significant correlation through univariate regression (ULR) for thyroid and hypopharyngeal tumors did not persist in multivariate analysis (MLR). Moreover, there was no significant association found for laryngeal tumors in both ULR and MLR, suggesting that HCMV may not confer protection in these cancers. As for the lack of connection between HCMV infection and malignant tumors of the larynx, our results are supported by the findings of Polz-Gruszka et al. conducted in 2015 [29], as well as and Strauss et al. conducted from 1995 to 2006 [30], both of which utilized the PCR method to investigate the relationship between HCMV and laryngeal neoplasms but failed to identify a significant correlation. Conversely, Schindele et al. reported findings suggestive of an oncomodulatory effect of HCMV, especially in co-infection with other viruses, though without a clear association with laryngeal malignancy [31]. Additionally, a 2015 case report documented HCMV presence in a laryngeal lymphoma patient, while similar findings were described in an immunodeficient patient elsewhere; however, neither have conclusively established a connection of cause and effect [32,33].

Initially, our research indicated a protective effect of HCMV in hypopharyngeal tumors, though subsequent multivariate analysis rendered this result inconsequential. In an immunohistochemical study conducted in 1993 on 46 patients with Kaposi’s sarcoma of the head and neck region, including the hypopharynx, Riederer et al. did not detect HCMV in any of the patients [34]. However, we wish to underscore that lack of viruses in a given tissue does not definitively exclude the effect of HCMV—pro-oncogenic or otherwise—as a hit-and-run infection modality is always a possibility.

Literature does not abound with studies inquiring into the interplay between thyroid cancer and HCMV. In the case of malignant thyroid tumors, our study, alongside findings from Kalavari et al., Huang et al., and Almeida et al., did not demonstrate a link between HCMV and thyroid malignancies [39,40,71]. Conversely, Saeed et al. in their 2021 paper detected HCMV in a notable percentage of patients using in situ hybridization, but again without a clear correlation [72]. HCMV co-infection in an immunocompromised patient with Kaposi’s sarcoma in the thyroid gland was noted in a case report [41].

In contrast to other H&N region tumors, a significant correlation has emerged between HCMV and nasopharyngeal tumors, where HCMV is positively associated with these cancers. The pro-tumor effect persists even after conducting multivariate analysis, implying that HCMV could be a potential oncogenic agent in said malignancies. Our results are related with those reported by Ahmed et al. and Chan et al., both of who have detected HCMV in a significant proportion of patients with nasopharyngeal cancer [35,36]. However, it is worthwhile noting that a 1983 study found no association between HCMV infection and nasopharyngeal malignancy [37].

Although studies investigating the etiological role of HCMV in nasopharyngeal malignant dyscrasia are few, HCMV has been implicated in a number of other tumors. Its potential oncogenicity has been theorized extensively [45,73,74], with suggestions that the virus might play a role in over 90% of the most commonly occurring tumors [75]. HCMV could potentially heighten the cell’s vulnerability to cancer through disrupting pathways related to the cell cycle, angiogenesis, cell invasion, apoptosis, and the host’s immune response [76,77]. In a study of oral squamous cell carcinoma, Polz-Gruszka et al. detected HCMV DNA in fresh-frozen tumor tissue from 10% of patients [29]. The pathogen has been described as a risk factor for glioma, neuroblastoma, as well as breast cancer [78,79,80,81,82,83]. In individuals with head and neck malignancies, the presence of HCMV antibodies did not directly affect survival. Nevertheless, the authors suggest that elevated antibody levels and the presence of an active HCMV in the tumor environment might be associated with poorer outcomes [84].

The virus has been suggested to contribute to the development of malignant gliomas [77,85,86,87,88,89,90,91,92,93,94], particularly the highly aggressive glioblastoma multiforme (GBM) [76,95,96]. At the very least, HCMV seems to play a role in oncomodulation [97], as it supports cancer growth and longevity [98,99,100], while enhancing its malignant potential by increasingly inducing a more neoplastic phenotype [85,86,99].

It is noteworthy that certain high-risk HCMV strains have been implicated in playing a catalytic role in the transformation of primary cells [3,6,98,101]. Notably, Cobbs observed that HCMV not only possesses the ability to modify epithelial cells but is also involved in the epithelial-to-mesenchymal (EMT) transformation in tumor cells, where in a vice versa effect was also observed. It is crucial to highlight that EMT has been proposed as the causative factor for deranged cellular polarity, loss of cell-to-cell adhesion, and transfiguration of the cellular cytoskeletal structure [102]. This process facilitates a critical role in tumor progression and is considered a primary target of interest in anticancer therapy [103]. Furthermore, HCMV has the ability to induce sequestration and deactivate p53, deepening our understanding of the virus’s purported pro-tumor properties [104].

The evidence from this and other mentioned studies raises a significant question: why does HCMV appear to be linked to oncoprotection in some cases, pro-tumorigenic in others, with no connection to the rest? The underlying cause for this observed inconsistency across the spectrum of these tumors remains unclear. It might be potentially associated with a diminished viral affinity for these specific histological environments. As for the purported oncopreventive faculty of HCMV, it is intriguing to note that this association statistically holds true from a global vantage point, and across many tumor histologies [14]. Namely, as the prevalence of HCMV increases, there is a corresponding decrease in cancer incidence. Conversely, HDI exhibits a co-incremental relationship with tumor pervasiveness. The rise in HDI in economically prosperous communities aligns with the adoption of improved hygiene, sanitation, education, and overall better health practices compared to less prosperous counterparts. This trend may lead to reduced transmission and seroprevalence of HCMV, thereby diminishing its potential oncopreventive impact. Such a connection further supports the comprehensive anti-cancer characteristics of HCMV infection observed across various tumor types globally [14].

## 5. Conclusions

The ongoing debate regarding the true nature of HCMV in cancer persists, and a comprehensive understanding of the interaction between H&N region tumors and the pathogen remains elusive. The challenge arises from conflicting findings; while some studies suggest a pro-tumor role of HCMV, other accumulating evidence indicates an overall anti-cancer effect. In this context, we present new data from a global perspective, revealing that the influence of HCMV varies depending on tumor histology. Our findings suggest a potential oncoprotective role of HCMV in salivary gland, lip/oral tissues, and the oropharynx. Conversely, the virus exerts an opposing influence in nasopharyngeal cancer. Importantly, there appears to be no discernible connection between HCMV and neoplasia of the hypopharynx, larynx, and thyroid.

The diverse effects of HCMV on various tissues observed in this study may be attributed to several factors. Different cell types could respond to HCMV in distinct ways. Additionally, more virulent strains of the pathogen may have a greater impact on specific tissues, such as the nasopharynx. Discrepancies in tumor incidence across countries with different socioeconomic statuses may be influenced by global variations in HCMV genotype distribution, although this may not explain the histologically-specific aspects of oncoprotection/oncogenesis. The role of population-specific genetic variations in these findings is currently unknown.

Variations in detection methods among studies that either support or oppose tumorigenesis may complicate comparisons among patient cohorts. Despite many investigations failing to identify HCMV in tumors or correlate it with cancer, this does not rule out the possibility of the hit-and-run hypothesis of neoplastic formation. Furthermore, HCMV may exert its influence in utero, predisposing a child to oncogenesis, or a primoinfection later in life may confer protection against malignancy.

It is crucial to note that correlation does not imply causation. Our global-scale results, while advantageous, should be interpreted in the context of detailed in vitro studies and comprehensive molecular analyses from other research that complements our investigation. The use of serological analyses in our study, suitable for manipulating large populations, was preferred over PCR, which may be suboptimal as HCMV reactivates infrequently and could prove elusive when tracked in this manner. We have considered various factors influencing HCMV seroprevalence and used regression analyses to account for potential confounders. However, we caution against drawing premature conclusions and suggest that our results be viewed as another stepping stone in understanding the true role of HCMV in tumors. More extensive molecular studies exploring host-pathogen interactions, along with large multicenter studies and broad epidemiological inquiries, are essential to validate our findings.

## Figures and Tables

**Figure 1 biomedicines-12-00872-f001:**
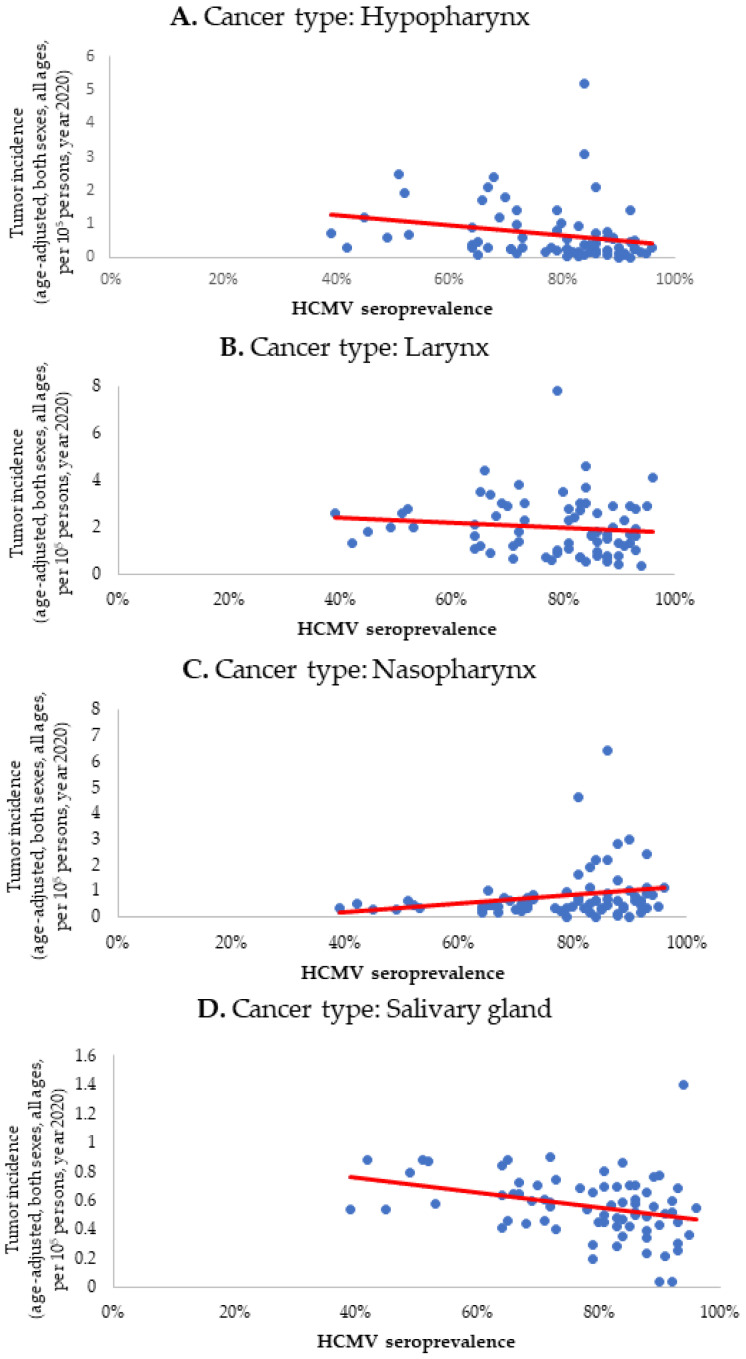
(**A**–**G**). The scatter-dot graphs represent the relation between HCMV seroprevalence from 73 countries (x-axis) and corresponding age-adjusted tumor incidences (per 100,000 persons) for both sexes and the 0–85+ age range for the year 2020 (y-axis; source: GLOBOCAN [16]). The age-adjusting compensates for tumor incidence differences between specific age groups. Analysis was carried out via Spearman correlation, with the coefficients shown in Table 2. Although the association is found relevant for each neoplasm, it is worthwhile noting that after multivariate regression analysis, the statistical significance remains only for tumors of the oropharynx, lip/oral cavity, salivary glands and nasopharynx.

**Table 1 biomedicines-12-00872-t001:** Details of the methodology employed in the PubMed^®^ searches, along with corresponding results.

Keywords	Filter	Result No.	Relevant Results [Reference No.]	Period (Years)
CMV, laryng*	Title	2	1 [29]	2005–2024
cytomegalov*, laryng*	Title	168	5 [29,30,31,32,33]	1960–2024
HHV5, laryng*	Title	0	0	N/A
HHV-5, laryng*	Title	0	0	N/A
CMV, laryn*	Title	2	1 [29]	2005–2024
Laryn*, cytomegalov*	Title	13	2 [31,33]	1981–2024
Laryn*, HHV5	Title	0	0	N/A
Laryn*, HHV-5	Title	0	0	N/A
Hypophary*, CMV	Title	6	1 [34]	1993–2024
Hypophary*, cytomegalov*	Title	8	0	1990–2024
Hypophary*, HHV5	Title	0	0	N/A
Hypophary*, HHV-5	Title	0	0	1990–2024
Nasopharyn*, CMV	Title	2	0	1999–2024
Nasopharyn*, cytomegalov*	Title	5	4 [35,36,37,38]	1979–2024
Nasopharyn*, HHV-5	Title	0	0	N/A
Nasopharyn*, HHV5	Title	0	0	N/A
Thyroi*, CMV	Title	3	0	2005–2024
Thyroi*, cytomegalov*	Title	13	4 [39,40,41,42]	1976–2024
Thyroi*, HHV5	Title	0	0	N/A
Thyroi*, HHV-5	Title	0	0	N/A
CMV, saliva*	Title	25	0	1994–2004
Cytomegalov*, saliva*	Title	131	8 [43,44,45,46,47,48,49]	1963–2024
HHV5, saliva*	Title	0	0	N/A
HHV-5, saliva*	Title	0	0	N/A
CMV, lip	Title	0	0	N/A
Cytomegalov*, lip	Title	1	0	2017
HHV5, lip	Title	0	0	N/A
HHV-5, lip	Title	0	0	N/A
Cytomegalov*, oral*	Title	48	1 [50]	2014–2024
CMV, oral*	Title	5	1 [51]	2014–2024
HHV5, oral*	Title	0	0	N/A
HHV-5, oral*	Title	0	0	N/A

**Table 2 biomedicines-12-00872-t002:** Predominant tumors of the head-and-neck region as documented by the World Health Organization’s GLOBOCAN, along with their connection to global CMV prevalence, analyzed with correlation and regression analysis.

Tumor/Localization	Correlation Analysis	Univariate Linear Regression Analysis			Multivariate Linear Regression Analysis ^a^		
	Spearman’s ρ	Stand. Coeff. β	95% CI	*p*-Value	Stand. Coeff. β	95% CI	*p*-Value *
Oropharynx	−0.651 *	−0.635	−5.817–−3.217	<0.001 *	−0.533	−5.473–−2.137	<0.001 *
Lip/Oral cavity	−0.551 *	−0.367	−8.984–−2.243	0.001 *	−0.368	−10.009–−1.273	0.012 *
Thyroid	−0.532 *	−0.485	−21.293–−8.557	<0.001 *	−0.138	−11.466–2.901	0.238
Hypopharynx	−0.377 *	−0.236	−2.963–−0.039	0.044 *	−0.261	−3.566–0.232	0.084
Salivary glands	−0.350 *	−0.320	−0.874–−0.154	0.006 *	−0.440	−1.165–−0.248	0.003 *
Nasopharynx	0.266 *	0.227	−0.023–3.451	0.053	0.338	0.337– 4.804	0.025 *
Larynx	−0.165	−0.117	−3.241–1.089	0.325	0.016	0.337– 4.804	0.913

^a^ Adjusted for country-specific human development index (HDI); Stand. Coeff. β—Standardized coefficients β; 95% CI—95% Confidence Interval; * Asterisks denote statistically significant values; significant at 0.05.

## Data Availability

All data are either publicly accessible or available from the corresponding author upon reasonable request.

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
