# Peer review of "Friend or Foe? Exploring the Role of Cytomegalovirus (HCMV) Infection in Head and Neck Tumors"

_biomedicines, 2024, doi:10.3390/biomedicines12040872_

Round 1

Reviewer 1 Report

Comments and Suggestions for Authors

Comment on “Friend or Foe? Exploring the Role of Cytomegalovirus (CMV) Infection in Head and Neck Tumors”:

If the authors want to look for a relationship between CMV and head and neck tumors, you have to formulate an appropriate hypothesis. Another alternative is to carry out a large systematic review on the impact of CMV on head and neck cancers.

Thank you so much

Line 67. Is the reference 50 [50]?

In the description of the results, comments that correspond to the discussion should be avoided, such as

Line 131: This observation suggests a plausible protective influence of the virus against the aforementioned neoplastic conditions on a global scale.

Line 145: This finding contradicts the apparent correlation observed between CMV prevalence and thyroid and hypopharynx tumor incidence on ULR analysis

Reviewer 2 Report

Comments and Suggestions for Authors

Trivic et al discuss the analysis of the relationship between head-and-neck tumors and CMV infection across 73 countries. Their findings suggest that CMV may have both protective and oncogenic effects depending on the tissue. Indeed, HCMV was found to have pro-oncogenic effects in patients with nasopharyngeal carcinoma, but an inverse (anti-tumor) association with tumors of the lip/oral region and salivary glands. While initially noted as protective for thyroid neoplasia and hypopharyngeal tumors, this connection did not hold after multivariate regression analysis. No association was found between laryngeal cancer and CMV infection.

Overall, the experiments seemed to be run correctly, and the analysis looks right. The topic is relevant and has not been deeply investigated in the literature so far. As such, new findings are required. The manuscript is well-written and organized. The tables are adequate and are of help to follow the manuscript.

However, some minor improvements are required to allow the publication of the work:

-          Human cytomegalovirus should be abbreviated as “HCMV”, not “CMV” in the title and throughout the manuscript;

-          Abstract:

“Conversely, a number of studies report on possible anti-tumor properties of the virus, which are recently being investigated for the purposes of anti-cancer treatment”. It is recommended to briefly mention the underlying principles (drugs? or viro-oncolytic therapy?) upon which these treatments are based.

“..more in-depth molecular analyses from real-world evidence…”: this sentence is a little bit elusive, what do the authors mean with the term “real-world”?

-          Line 58: head and neck should be abbreviated as “H&N” not “HAN”

Reviewer 3 Report

Comments and Suggestions for Authors

The authors aimed to provide a more comprehensive understanding of the correlation between this virus and specific head and neck (HAN) tumors on a global scale. Intriguingly, CMV was found to be pro-oncogenic in patients with nasopharyngeal carcinoma; in contrast, the virus exhibited an inverse (i.e., anti-tumor) association with tumors of the lip/oral region and the salivary glands. The manuscript is very informative and suggests many potential mechanisms for future exploration. However, the author needs to revise several points before the manuscript can be accepted.

Revision Points:

1. Table 1 includes similar keywords but listed independently. The authors need to explain the reason for this distinction or revise them accordingly.

2. The authors used a complex calculation methodology. How can other readers reexamine the manuscript's results? The authors need to provide a flowchart of the calculation methodology and/or include detailed calculation formulas as supplementary material.

Round 2

Reviewer 1 Report

Comments and Suggestions for Authors

Thank you  for the answers